# Improved Multimodal Tumor Necrosis Imaging with IRDye800CW-DOTA Conjugated to an Albumin-Binding Domain

**DOI:** 10.3390/cancers14040861

**Published:** 2022-02-09

**Authors:** Marcus C. M. Stroet, Erik de Blois, Marion de Jong, Yann Seimbille, Laura Mezzanotte, Clemens W. G. M. Löwik, Kranthi M. Panth

**Affiliations:** 1Erasmus MC, Department of Radiology & Nuclear Medicine, University Medical Center Rotterdam, 3015 GD Rotterdam, The Netherlands; m.stroet@erasmusmc.nl (M.C.M.S.); r.deblois@erasmusmc.nl (E.d.B.); y.seimbille@erasmusmc.nl (Y.S.); l.mezzanotte@erasmusmc.nl (L.M.); 2Erasmus MC, Department of Molecular Genetics, University Medical Center Rotterdam, 3015 GD Rotterdam, The Netherlands; 3Life Sciences Division, TRIUMF, Vancouver, BC V6T 2A3, Canada; 4CHUV Department of Oncology, University of Lausanne, CH-1066 Lausanne, Switzerland

**Keywords:** necrosis-avid contrast agent, cyanines, multimodal imaging, therapy efficacy

## Abstract

**Simple Summary:**

Anti-tumor treatment efficacy is determined by tumor shrinkage, which takes valuable time to become apparent and poses a risk of unnecessary treatment with severe side effects. Therefore, there is an unmet need for more reliable and specific methods to monitor treatment efficacy. We explore radiolabeled cyanines for imaging tumor necrosis as a unique marker for therapy efficacy. Moreover, spontaneous tumor necrosis is a hallmark for aggressively growing tumor types with poor prognosis. We improved the binding properties of a previously reported necrosis-avid contrast agent (NACA) and successfully detected spontaneous and therapy-induced tumor necrosis in mice using radioactivity and fluorescence imaging modalities. This NACA may pave the way to in vivo detection of tumor necrosis for early-stage determination of tumor aggressiveness and therapy efficacy.

**Abstract:**

Purpose: To assess our improved NACA for the detection of tumor necrosis. Methods: We increased the blood circulation time of our NACA by adding an albumin-binding domain to the molecular structure. We tested the necrosis avidity on dead or alive cultured cells and performed SPECT and fluorescence imaging of both spontaneous and treatment-induced necrosis in murine breast cancer models. We simultaneously recorded [^18^F]FDG-PET and bioluminescence images for complementary detection of tumor viability. Results: We generated two albumin-binding IRDye800CW derivatives which were labeled with indium-111 with high radiochemical purity. Surprisingly, both albumin-binding NACAs had >10x higher in vitro binding towards dead cells. We selected [^111^In]**3** for in vivo experiments which showed higher dead cell binding in vitro and in vivo stability. The doxorubicin-treated tumors showed increased [^111^In]**3**-uptake (1.74 ± 0.08%ID/g after saline treatment, 2.25 ± 0.16%ID/g after doxorubicin treatment, *p* = 0.044) and decreased [^18^F]FDG-uptake (3.02 ± 0.51%ID/g after saline treatment, 1.79 ± 0.11%ID/g after doxorubicin treatment, *p* = 0.040), indicating therapy efficacy. Moreover, we detected increased [^111^In]**3-**uptake and tumor necrosis in more rapidly growing EMT6 tumors. Conclusions: Our albumin-binding NACA based on IRDye800CW facilitates tumor-necrosis imaging for assessment of therapy efficacy and aggressiveness in solid tumors using both fluorescence and SPECT imaging.

## 1. Introduction

Necrotic forms of cell death are associated with the severity of several diseases, including myocardial infarction, stroke, sepsis, pancreatitis, and solid tumors where it plays an important role in tumor proliferation and invasion [1]. Most tumors spontaneously develop necrotic tissue due to underdeveloped and less functional vasculature that is incapable to keep up with the rapid tumor growth, which is more profound in aggressive cancers with poor prognosis. Hence, detecting tumor necrosis may serve as a tool to stage tumor aggressiveness [2]. Moreover, effective chemotherapy induces cell death in the tumor. Even though chemotherapies can kill tumor cells via several forms of cell death, within a tumor it is practically impossible to discriminate these forms (i.e., incidental necrosis, secondary necrosis, and all forms of regulated necrosis) [3,4,5,6]. Therefore, early detection of increased cell death that occurs after chemotherapy is an unmet need. It is clear that detection of necrosis provides essential diagnostic and prognostic information but there are no robust necrosis probes available for clinic applications.

In current clinical practice, therapy efficacy against solid tumors is determined by tumor volume reduction in CT or MRI scans [7,8], or by changes in tumoral metabolic activity detected by [^18^F]FDG-PET [9]. However, these methodologies have considerable limitations: it takes valuable time and often multiple lines of treatment for reduction in tumor volume to become apparent and the image interpretation may be prone to human error [10]. Moreover, [^18^F]FDG-PET imaging of small or less glycolytic lesions is troublesome [11] and tracer uptake can be confounding with parallel disease processes [12]. All of this could lead to excessive exposure of the patient to ineffective treatments with severe side effects. Hence, there is an unmet need for fast and reliable methods to determine therapeutic efficacy. One such methodology would be detecting therapy-induced cell death since most anti-tumor therapies are designed to kill tumor cells [1].

We described the ability of NIR fluorescent cyanine dyes to bind to cells that lost their membrane integrity to image cell death [4,13,14,15]. IRDye800CW is a widely applied fluorescent dye with biocompatible near-infrared (NIR) fluorescent properties (absorption: 774 nm, emission: 789 nm). IRDye800CW has a strong avidity for dead cells even when bound to small molecules or peptides [13,16]. We recently radiolabeled the cyanine IRDye800CW conjugated to DOTA with SPECT-isotope indium-111 for imaging of tumor necrosis in a mouse model. This tracer targeted necrotic tumors with high tumor-to-background ratios but suffered rapid renal excretion [15].

Adding an albumin-binding domain (ABD) in the molecular structure of our radiotracer could extend its blood circulation time. Albumin is the most abundant blood protein, which binds and transports molecules throughout the body [17]. Moderate binding to albumin increases the possibility of a tracer reaching its target tissue by limiting the metabolic degradation and renal filtration [18]. This relatively simple approach has ample examples of nuclear imaging agents with improved target uptake [19,20].

We explored dead cell binding of radiolabeled IRDye800CW derivatives with an albumin-binding domain and compared them to our earlier described [^111^In]In-DOTA-PEG_4_-IRDye800CW ([^111^In]**1**, Figure 1) [15]. We propose two constructs containing a DOTA-chelator for indium-111 labeling, the dead-cell-binding IRDye800CW, and a 4-(*p*-iodophenyl)butyramide moiety as an albumin-binding domain [21]. The constructs are designed with lysine residues as anchor points of the different molecular components ([^111^In]**2** and [^111^In]**3**, Figure 1), and we investigated if these compounds can be used to determine the effect of chemotherapy on solid tumors, using murine breast cancer models. Moreover, we investigated if we could use these compounds to determine tumor aggressiveness by comparing uptake in slow and fast proliferating tumor models, characterized by different levels of necrosis.

## 2. Materials and Methods

### 2.1. Materials

The 4T1-Luc2 cells were purchased from PerkinElmer (Boston, MA, USA). MCF7 and EMT6 cells were purchased at ATCC catalog number HTB-22 and CRL-2755, respectively. Both cell lines were lentivirally transduced with the construct pCDH-EF1-Luc2-copGFP as previously described [22].

Cell culture media was obtained from Sigma-Aldrich (St. Louis, MO, USA) or Gibco Life Technologies (Waltham, MA, USA). Reagents were purchased from Sigma-Aldrich (St. Louis, MO, USA) unless stated otherwise. Solvents were purchased from Honeywell Riedel-de-Haën™ (Seelze, Germany). IRDye800CW *N*-hydroxysuccinimide (NHS) ester was purchased from Westburg BV (Leusden, The Netherlands). [^111^In]InCl_3_ from Mallinckrodt BV (Petten, The Netherlands). All Fmoc-amino acids were purchased from Iris Biotech (Marktredwitz, Germany). Cell viability assay (MTS) was performed with CellTiter 96^®^ Aqueous One Solution Cell Proliferation Assay (Promega, Madison, WI, USA). Autoradiography was performed using super-resolution phosphor screens and a Cyclone^®^ Plus system (Perkin Elmer, Waltham, MA, USA). The 800 nm fluorescence imaging was performed on an Odyssey flatbed scanner system (800 nm channel, laser intensity 2.0; Li-Cor, Lincoln, NE, USA). FLI/BLI and SPECT/CT images were obtained with a VECTor^5^OI/CT (MILabs, Delft, The Netherlands). SPECT images were recorded with a high-sensitivity 3.0 mm pinhole mouse collimator, and SPECT/PET images were recorded with a high-energy general-purpose 1.6 mm pinhole mouse collimator. The images were analyzed with Pi-Mod (version 3.901) and visualized in VivoQuant (version 2.50, patch 3). The activity was accurately quantified on a Wizard 3″, 1480 γ-counter (PerkinElmer). Cell viability staining for microscopy was performed with Live/Dead Cell Double Staining Kit from Sigma-Aldrich. Histochemical dead cell staining was performed using the DeadEnd™ Colorimetric TUNEL System (Promega, Madison, WI, USA) and subsequent imaging on a NanoZoomer 2.0HT digital slide scanner (Hamamatsu, Hamamatsu City, Japan). Fluorescent TUNEL dead cell staining was performed using the In Situ Cell Death Detection Kit, Fluorescein (Roche Diagnostics GmbH, Mannheim, Germany). Stitched fluorescence microscopy images of the stained sections were acquired on a fluorescence microscope (TriPath Imaging, Burlington, NC, USA) equipped with an Axiocam MRm (Zeiss, Oberkochen, Germany) and an Axiocam 208 color (Zeiss) using a magnification of 40×.

### 2.2. Precursor Synthesis and Radiolabeling

Preparation of [^111^In]**1** is described elsewhere [15]. The synthesis of precursors **2** and **3** are described in Appendix A and the radiolabeling procedures for radiotracers [^111^In]**2** and [^111^In]**3** can be found in Appendix A.

### 2.3. Determination of n-Octanol/PBS Distribution Coefficient (LogD_7.4_)

The [^111^In]**1**, [^111^In]**2**, and [^111^In]**3** were labeled at a molar activity of 10 MBq/nmol and 100 kBq was added to *n*-octanol (500 µL) and PBS (500 µL). After vigorous vortexing, the vial was centrifuged at 16.1 × 1000 *g* for 3 min. The octanol phase was carefully pipetted into a new vial and both layers were centrifuged again. From both the organic and the aqueous layers, samples (10 µL) were taken out and counted for activity on a γ-counter. LogD values were calculated as follows: Log ([NACA]*_n_*_-octanol_/[NACA]_PBS_). These experiments were performed as triplicates.

### 2.4. In Vitro Protein Binding

Protein binding of the NACAs was determined as reported elsewhere [23]. The NACAs were labeled with indium-111 (50 MBq/nmol), added to mouse serum (1 MBq/500 µL), and incubated at 37 °C for 30 min. Four portions of 10 µL were taken and counted on a γ-counter. The remaining serum was transferred to a centifree ultrafiltration device (4104 centrifugal filter units; Millipore, 30 kDa nominal molecular weight limit, methylcellulose micro partition membranes equilibrated once with 5% *v*/*v* Tween-80 in phosphate-buffered saline (PBS) and then trice with PBS) and centrifuged at 7.0 × 1000 *g* for 40 min at room temperature. Four portions of 10 µL from the filtrates were counted for radioactivity in a γ-counter. The amount of plasma-bound radioligand was calculated as the fraction between the radioactivity measured in the filtrate portions relative to the corresponding loading solution (set to 100%).

### 2.5. In Vitro Dead/Alive Cell Binding Assay

The 4T1-Luc2 cells were cultured in RPMI-1640 medium containing 10% fetal bovine, penicillin (100 U/mL), and streptomycin (100 mg/mL) 37 °C under a humidified atmosphere with 5% CO_2_. The cells were seeded in 24 well plates (1 × 10^5^ cells per well). The next day, the cells were killed with EtOH (70%, 50 μL) or kept alive and incubated for 30 min at 37 °C with 100 nM of the NACAs in culturing medium. The plate was then imaged on an Odyssey (Li-Cor, 800 nm laser intensity 5.0).

Similarly, these killed and alive cells were incubated with the indium-111 labeled NACAs (100 nM, 2.5 MBq/nmol) in culturing medium. The cells were washed with PBS and collected with NaOH (1.0 M) for γ-counting. These experiments were performed as triplicates.

### 2.6. Microscopy

The 4T1-Luc2 cells were seeded on 10 mm diameter coverslips in 24 well plates (10^5^ cells per well). The cells were kept alive or killed with 50 µL EtOH (70%) and incubated in culturing medium with or without unlabeled **3** (100 nM) for 30 min at 37 °C. The cells were gently washed with PBS three times and incubated with Calcein-AM (CAM, 2 µM) and propidium iodide (PI, 4 µM) in PBS for 30 min at 37 °C [24]. Then, the cells were fixated with PFA (4% in PBS, 10 min, RT) and nuclei were stained with DAPI (Sigma-Aldrich, 1:1000 in PBS, 5 min). After final washing with PBS, the coverslips were removed from the well plate and mounted on a microscope slide for fluorescence microscope imaging at 20× magnification.

### 2.7. Animals

Female BALB/cAnNRj-nude mice (six to eight weeks old) were housed in ventilated cages in groups of four to six mice and were provided standard laboratory animal food pellets and water ad libitum. One week after arrival, tumor cells were injected bilaterally on the shoulders. Tumor growth was monitored three days per week with a caliper. All groups consisted of four mice with bilateral tumors.

### 2.8. In Vivo Stability

Mice were injected with [^111^In]**2** or [^111^In]**3** (20 MBq, 2 nmol, in 200 µL PBS) into the tail vein. After an hour, the mice were sacrificed by cervical dislocation and blood and urine were collected, which were frozen in liquid nitrogen directly after extraction. For analysis, the blood and urine were thawed and diluted with acetonitrile (*v*/*v*, 1:1) for protein precipitation and centrifuged at 16.1 × 1000 *g* for 15 min at room temperature. The supernatant was collected and analyzed by radio-HPLC (Method-D).

### 2.9. FLI, BLI, and SPECT/CT Tumor Necrosis Imaging, Ideal Timepoint Determination

The 4T1-Luc2 cells (1.0 × 10^5^) suspended in 15 µL PBS were inoculated bilaterally in 16 mice in four groups of four animals each. Nine days later, the tumor-bearing mice were injected with [^111^In]**3** (20 MBq, 2 nmol, in 200 µL PBS) into the tail vein, after which imaging was performed at timepoints 6, 24, 48, and 72 h post-injection (p.i.) under isoflurane anesthesia (4% induction, 1.5 to 2% maintenance in 100% O_2_), whilst maintaining the body temperature constant. First, FLI imaging was performed with a 730 nm excitation and 760 nm emission filter for 400 ms with 4 × 4 binning. Then, the mice received an intraperitoneal injection of D-luciferin (150 mg/kg) maintaining the position of the mice in the bed unaltered, and BLI imaging was performed with an open filter for 400 ms with 4 × 4 binning. Full body (head to tail base) SPECT images of the 5 cm axial field of view were obtained over a total scan time of 30 min, followed by a 2 min full-body CT. After imaging, the mice were euthanized for ex vivo analysis.

### 2.10. FLI, BLI, and PET/SPECT/CT Tumor Necrosis Imaging Procedure

The quadruple modality imaging procedure is illustrated in Appendix A. The tumor-bearing mice were injected with [^111^In]**3** (20 MBq, 2 nmol) in 200 µL PBS into the tail vein. The following day, the mice fasted for four hours before they were injected with [^18^F]FDG (10 MBq, 200 µL) in PBS. At 24 h p.i. of [^111^In]**3** and 1 h p.i. of [^18^F]FDG, FLI imaging was performed with a 730 nm excitation and 760 nm emission filter for 400 ms with 4 × 4 binning. Then, the mice received an intraperitoneal injection of D-luciferin (150 mg/kg) maintaining the position of the mice in the bed unaltered. During the onset of the bioluminescence signal, PET/SPECT imaging was performed of the tumors with a 2 cm axial field of view over a total scan time of 30 min, followed by a 2 min full-body CT. Finally, BLI imaging was performed with an open filter for 400 ms with 4 × 4 binning. Imaging was performed under isoflurane anesthesia whilst maintaining the body temperature constant.

### 2.11. Doxorubicin Treatment-Induced Necrosis in MCF7-Luc2

Six mice were inoculated bilaterally with MCF7-Luc2 cells (3.0 × 10^6^), suspended in 30 µL PBS:Matrigel (1:1; *v*/*v*). On days 14, 16, and 18 after inoculation, the tumor-bearing mice received either doxorubicin (5 mg/kg, Janssen Pharmaceuticals, Beerse, Belgium) or a saline solution via i.v. injection. On days 21 and 22 after inoculation, the mice underwent the FLI, BLI, and PET/SPECT/CT imaging procedure (vide supra).

### 2.12. FLI, BLI, and PET/SPECT/CT Necrosis Imaging in EMT6-Luc2 Tumors

Three mice were inoculated bilaterally with EMT6-Luc2 cells (5.0 × 10^5^) suspended in 15 µL PBS. On days 9 and 10 after inoculation, the tumor-bearing mice were subjected to the FLI, BLI, and PET/SPECT/CT imaging procedure (vide supra).

### 2.13. SPECT/PET/CT Image Processing

Acquired SPECT and PET images were reconstructed using SR-OSEM with 9 iterations and 128 subsets on a 36 × 36 × 35 mm matrix with 0.80 mm isotropic voxels. For the PET images, a decay correction of 109.7 min was applied. The PET and SPECT images were superposed with CT. The images were further analyzed in Pi-Mod. Regions of interest were manually drawn around tumors in the CT images. The uptake values in the regions of interests were corrected for volume of the region of interest, injected dose, and time of injection to obtain the uptake values as the percentile of injected dose per volume (%ID/mL).

### 2.14. Ex Vivo Analysis

After the final imaging experiments, the mice were sacrificed by cervical dislocation, tumors and organs were collected, weighted, and activity was accurately quantified by γ-counting to determine radioactivity uptake in the percentage of injected dose per gram (%ID/g). After γ-counting, the tumors were frozen in liquid nitrogen and stored at −80 °C overnight. Adjacent cryosections (10 µm) were prepared of the tumor centers for autoradiography. The sections were stored at −20 °C for the radioactivity to decay, before 800 nm fluorescence imaging and histochemical dead cell staining.

### 2.15. Statistics

All data are expressed as the mean ± SD unless stated otherwise. Outliers were identified and excluded by a Q-test. *p* values of <0.05 were considered significant. Significance was determined with a two-tailed t-test in Microsoft Excel 2016. Graphs were prepared in Microsoft Excel 2016 or GraphPad Prism 5.

## 3. Results

### 3.1. Precursor Synthesis and Radiolabeling

We designed precursor **2** with a double lysine backbone to conjugate the DOTA-chelator, IRDye800CW, and the ABD on a single molecule. Moreover, this configuration allowed the synthesis of the precursor to be performed predominantly by solid-phase peptide synthesis. The synthesis began with immobilizing Fmoc-L-Lys (Boc)-OH onto the Rink amide MBHA resin. The Fmoc group was removed under basic conditions before conjugation of ivDde-L-Lys (Fmoc)-OH to the first lysine (Figure 1, steps a–c). The Fmoc deprotection was followed by the conjugation of the albumin binding 4- (*p*-iodophenyl)butyric acid (steps d,e). Subsequently, the ivDde-protecting group was selectively removed with hydrazine, and DOTA-tris (*t*Bu)ester was conjugated to the N-terminus, while the Boc-protecting group on the side chain amine of the first lysine residue was preserved. The following step was the concomitant cleavage from the resin and removal of the Boc and *t*Bu protecting groups (steps f–h). The last step of the synthesis was the conjugation of IRDye800CW *N*-hydroxysuccinimide (NHS) ester with the intermediate **4** to yield precursor **2**, which was purified by semi-preparative HPLC and characterized by HPLC and mass spectrometry. The overall yield was 3.1% over nine steps.

Next, precursor **3** was prepared with a single lysine backbone to incorporate the DOTA-chelator, the IRDye800CW, and the albumin-binding domain. The synthesis was initiated by the conjugation of Fmoc-L-Lys (Boc)-OH onto a 2-chlorotrityl resin, followed by Fmoc deprotection and insertion of the ABD at the N-terminal position (Figure 2, steps a–c). After this short SPPS sequence, the material was cleaved from the resin, preserving the Boc-protection group. The carboxylic acid **5** was coupled to *p*-xylylene diamine to obtain amine **6**, which was conjugated to DOTA (tris-*t*Bu)ester. The *t*Bu- and Boc-protecting groups were removed by treatment of **7** with TFA. The crude compound **8** was reacted with IRDye800CW NHS-ester to provide precursor **3**, which was purified by semi-preparative HPLC and characterized by HPLC and mass spectrometry. The overall yield was 7.1% over eight steps.

Precursors **2** and **3** were conveniently prepared by a standard solid-phase peptide synthesis approach and were prepared with >98% purity after semi-preparative HPLC. All analytical data of the intermediate and final compounds are presented in Appendix A). Both precursors were titrated and stored in 10 nmol aliquots at −20 °C [25]. Radiolabeling with indium-111 was performed at a molar activity of 10 MBq/nmol at which a radiochemical purity of >95% was reliably achieved (Appendix A). The radiochemical purities were rapidly determined by an iTLC method to streamline the radiolabeling [15].

### 3.2. Log D and Protein Binding

The logD value of our albumin-binding derivatives decreased hydrophilicity, compared to our previous necrosis tracer [^111^In]**1**. Including an ABD in the molecular structure reduced the hydrophilicity of the NACA by almost one order of magnitude. Unsurprisingly, the protein binding of the tracers significantly increased from 70% for [^111^In]**1** to 94–95% for the albumin binding NACAs (*p* < 0.0001, Table 1).

### 3.3. In Vitro Dead/Alive Cell Binding

Non-radioactive precursors were incubated with dead and alive 4T1-cells to detect the specific dead cell binding. As described before, IRDye800CW-based compound **1** specifically bound to dead cells and not to live cells [15]. Strikingly, the albumin-binding IRDye800CW derivatives (**2** and **3**) demonstrated a considerably stronger binding to the dead cells than compound **1** (Figure 2A,B). Intermediate **8** was isolated and radiolabeled to verify the role of the ABD on the dead cell binding. Compound **8** is corresponding to precursor **3**, lacking the dead-cell-binding IRDye800CW. The resulting [^111^In]**8** was included in the dead-cell-binding assay alongside radiolabeled compounds [^111^In]-**1**, [^111^In]**2,** and [^111^In]**3**. The dead cell binding of [^111^In]**2** and [^111^In]**3** was increased as compared to [^111^In]**1** (*p* = 8.72 × 10^−14^, and *p* = 4.98 × 10^−10^, respectively) while construct [^111^In]**8** did not show considerable dead cell binding (Figure 2C). The specific dead cell binding of precursor **3** was further confirmed by fluorescence microscopy.

### 3.4. Microscopy

A distinct NIR-fluorescent signal was observed from dead cells exposed to compound **3** (Figure 3). The viability of the cells was confirmed by CAM and PI, which stain for metabolic activity of live cells and exposed DNA of dead cells with disrupted cell membranes, respectively. The binding pattern of compound **3** is different than that of PI. Compound **3** binds to cytosolic structures, while PI binds exclusively to the DNA in the nuclei. Moreover, the NIR-fluorescence was undetectable in the cells incubated with culturing medium without compound **3**, which excluded the presence of autofluorescence. The NIR-fluorescent signal was predominantly observed in the cytoplasm but also partially in the nuclei.

### 3.5. In Vivo Stability

Six mice received an injection of [^111^In]**2** or [^111^In]**3** to test the in vivo stability of the constructs. At one hour after injection, blood and urine were collected from the animals and analyzed by radio-HPLC. Due to binding to blood proteins, not enough radioactivity could be obtained from the blood samples. Most [^111^In]**2** was decomposed after an hour, whereas [^111^In]**3** remained stable (>95%) at 1 h p.i. (Appendix A). For this reason, and the lower in vitro binding profile of [^111^In]**2**, [^111^In]**3** was selected for the following experiments.

### 3.6. In Vivo Necrosis Imaging

We explored in vivo necrosis imaging with [^111^In]**3** in 4T1-Luc2 tumor-bearing mice, a spontaneous necrotic tumor model. We injected [^111^In]**3** for imaging and performed biodistribution studies at different time points to determine the optimal time point after injection for imaging (Figure 4A). The radioactive tumor uptake of [^111^In]**3** in the SPECT images was 3.83 ± 0.19%ID/mL at 24 h p.i., which is higher than the tumor uptake of [^111^In]**1** (0.26 ± 0.05%ID/mL) [15]. The background signal was low enough to visualize the tumors on the SPECT scans at 24 h p.i., which was selected as the optimal time point for imaging.

The luciferase expression of the 4T1-Luc2 tumors facilitates BLI imaging of viable tumor cells. Resulting in necrotic tumors that appear on BLI images as a ring with a “dark” spot, due to a lower signal in the middle. The fluorescent signal localized in the parts of the tumors with reduced bioluminescence activity (Figure 4B). The same BLI and FLI signal pattern was observed in the freshly resected tumor (Figure 4C).

As expected, a distinctly higher blood concentration was observed as compared to the earlier described [^111^In]**1**, which does not contain an albumin-binding domain (Figure 4D, values in Appendix A). The blood half-live of [^111^In]**3** was with 9 h considerably longer than the one-hour half-life time of [^111^In]**1 [15]**. There is a delayed uptake pattern in the kidneys, which peaks around 24 h p.i. The bone marrow uptake seems to increase at 72 h p.i. but it is insignificant as compared to the uptake at 48 h p.i. (*p* = 0.07). The bone marrow uptake values should moreover be considered with care. The small amount of bone marrow that can be extracted (1–2 mg) introduces a considerable weighing error margin. We also noticed a distinct uptake and retention of the radioactivity in the tumors. Autoradiography, NIR fluorescence imaging, and TUNEL dead cell staining on adjacent tumor sections confirmed specific binding of the tracer to necrotic tumor regions (Figure 4E).

MCF7-Luc2 tumor-bearing mice were treated with doxorubicin to determine if therapy-induced necrosis could be detected with [^111^In]**3**. MCF7-Luc2 tumors proliferate less aggressively than 4T1-tumors and should, therefore, develop less spontaneous necrosis [26]. Representative four-modality imaging (PET/CT, SPECT/CT, FLI, and BLI) is presented in Figure 5A, showing two MCF7-Luc2 tumor-bearing mice that either received three injections of saline or doxorubicin. The quantified tumor uptake of [^18^F]FDG was reduced significantly in the doxorubicin-treated group (3.02 ± 0.51%ID/g after saline treatment, 1.79 ± 0.11%ID/g after doxorubicin treatment, *p* = 0.040), whereas the uptake of [^111^In]**3** was significantly higher (1.74 ± 0.08%ID/g after saline treatment, 2.25 ± 0.16%ID/g after doxorubicin treatment, *p* = 0.044, Figure 5B). Considerable [^111^In]**3**-uptake by tumors in the control group indicated the presence of intrinsic tumor necrosis. The uptake patterns within the tumors of [^111^In]**3** and [^18^F]FDG were heterogeneous and complementary to each other. Similarly, the FLI signal from [^111^In]**3** in the tumor centers was complementary to the BLI signal, mainly visible in the viable outer rim of the tumor. Ex vivo analysis revealed heterogeneous radioactivity and NIR-fluorescence uptake in the tumor tissue, overlapping with the TUNEL-positive regions, thereby confirming specific binding of [^111^In]**3** to dead cells in the tumor. Moreover, TUNEL dead cell staining confirmed the presence of tumor necrosis in the untreated MCF7-Luc2 tumors (Figure 5C).

Figure 6 shows the PET/CT and SPECT/CT images of EMT6-Luc2 tumor-bearing mice. Similar to Figure 5, heterogeneous and complementary tumor uptake patterns of [^18^F]FDG and [^111^In]**3** were observed. The [^111^In]**3** uptake is higher in the EMT6-Luc2 tumors than the control MCF7-Luc2 tumors (2.81 ± 0.21%ID/g in EMT6-Luc2 and 1.74 ± 0.08%ID/g in MCF7-Luc2, *p* = 0.00012), while the [^18^F]FDG is insignificantly different (4.13 ± 0.58%ID/g in EMT6-Luc2 and 3.02 ± 0.51%ID/g in MCF7-Luc2, *p* = 0.18). EMT6-Luc2 cells have a faster doubling rate than MCF7-Luc2 cells (12 h and 43 h, respectively), resulting in more aggressively growing tumors with more necrosis [27,28]. Again, NIR-fluorescence, autoradiography, and TUNEL dead cell staining on frozen tumor sections confirmed specific tumoral dead cell binding (Figure 6C).

## 4. Discussion

We previously showed that certain cyanines bind to cytosolic proteins of dead cells that lost membrane integrity [13]. We reported radiolabeled cyanine [^111^In]**1** as a necrosis-avid contrast agent (NACA). However, its rapid renal excretion limited the total tumor uptake [15]. Adding an ABD in the molecular structure is an attractive approach to increase the in vivo blood circulation time of nuclear contrast agents [20,21]. One of the most reported ABDs is the 4-(*p*-iodophenyl)butyrate moiety, which we selected for our NACA conjugates [29,30,31,32].

Although the primary aim was only to increase the blood circulation of the tracer, adding an ABD drastically improved in vitro dead cell binding as compared to compound **1**. This increased necrosis binding not only enabled detecting compound **3** with fluorescence microscopy but even in vivo detection of tumor necrosis with NIR-FLI, which was impossible with [^111^In]**1** due to its rapid excretion [15]. The absence of specific dead cell binding by [^111^In]**8** excluded any specific avidity for dead cells by the ABD itself (Figure 2). Furthermore, the total binding to both dead and alive cells by [^111^In]**8** was similar to the dead cell binding of [^111^In]**1**, indicating some degree of non-specific protein binding. Although it is unclear what directly causes the increased in vitro dead cell binding, a possible explanation is the increased lipophilicity and the protein binding of the albumin-binding derivatives. Kuo et al. reported a similar finding with a PSMA-targeting compound, conjugated with the same 4-(*p*-iodophenyl)butyrate moiety and had increased in vitro binding to PSMA-expressing cells [30]. Irrespective of the mode of action behind the dead cell binding of these cyanines, we observed specific avidity to dead cells and not to live cells.

We found a tumor-to-blood ratio of 2.50 ± 0.20 and a tumor-to-muscle ratio of 7.27 ± 0.34 at the 24 h p.i. timepoint, which we selected for imaging of the necrotic tumors. Similar to most albumin-binding nuclear contrast agents, the kidney uptake had a slow onset and peaked after 24 h p.i. (Figure 4) [20]. There is also a slow onset of uptake in the lymph nodes, which can be due to the gradual diffusion of albumin into the lymphatic system [32]. The notable kidney uptake and low uptake by liver, intestines, and spleen revealed that [^111^In]**3** was predominately excreted via the renal tract, which is likely due to the high hydrophilicity of the [^111^In]**3**. Although MCF7-Luc2 tumors can develop spontaneous necrosis [33], we measured an increased tumor uptake of [^111^In]**3** in doxorubicin-treated mice. The effect was even confirmed by the decreased [^18^F]FDG -uptake, which is a golden standard for clinical evaluation of therapy efficacy in solid tumors [34]. The distinctly different emission spectra from fluorine-18 and indium-111 allowed simultaneous imaging of the two radionuclides [35]. Metabolism tracer [^18^F]FDG coincides with the viable tumor regions with a low signal from [^111^In]**3**. Vice versa, [^111^In]**3** binds to the necrotic tumor regions with a low [^18^F]FDG-PET signal (Figure 5). [^18^F]FDG-uptake has often proved to be a reliable indicator for therapy efficacy assessment in many tumor types. However, necrosis detection may be a good and reliable alternative for efficacy assessment as [^18^F]FDG-uptake can be confounding with inflammation reactions and is not suitable for every tumor type [12]. Furthermore, measuring a decrease in [^18^F]FDG-uptake (signal) in small and/or low glycolytic active tumors is challenging.

Several imaging agents are under investigation for detecting cell death, with some of them reaching clinical trials. For instance, imaging agents [^18^F]ICMT-11 or Annexin V were tested clinically, targeting activated caspase-3 or externalized phosphoserines, respectively. However, these tracers suffered from low specificity and high abdominal uptake [1,36]. Therefore, many efforts are made to generate better contrast agents for imaging cell death. For instance, C2Am is a 16 kDa peptide that binds to late apoptotic and necrotic cells by targeting phosphoserines with greater signal-to-noise ratios as compared to annexin V. Szucs et al. demonstrated fluorescent and optoacoustic necrosis imaging using fluorescently labeled C2Am [37]. Besides, the same group reported earlier that C2Am can also be labeled with technetium-99m or indium-111 for SPECT imaging or with fluorine-18 for PET-imaging of therapy-induced tumor necrosis [38,39]. Necrosis may be a more specific target than apoptosis since apoptosis occurs abundantly for maintaining tissue homeostasis, whereas necrosis is absent in healthy persons [40]. IRDye800CW is a well-known fluorescent dye with little to no toxicity, which has already found many applications in the clinic [41,42]. The favorable biodistribution and necrosis avidity of [^111^In]**3** makes it a promising candidate for in vivo necrosis imaging in the clinic.

The comparison of MCF7-Luc2 to EMT6-Luc2 tumors in Figure 6 revealed the possibility to detect aggressiveness of the tumors. EMT6-Luc2 cells have 3.6 times faster in vitro doubling time than MCF7-Luc2 cells. In mice, this difference in tumor growth is more profound as EMT6 is a murine cell line and MCF7 is a human cell line, resulting in more aggressively growing tumors with more necrosis [27,28]. The EMT6-Luc2 tumors had significantly higher [^111^In]**3**-uptake but insignificantly higher [^18^F]FDG-uptake than the slower-growing MCF7-Luc2 tumors. Faster tumor growth requires higher glycolytic activity but the increased occurrence of tumor necrosis reduces the overall tumor [^18^F]FDG-uptake. The presence of tumor necrosis is recognized as a hallmark for rapidly proliferating tumors and with poor prognosis [5,43,44,45]. Moreover, the here-presented NACA can be a valuable tool for in vivo studies on the role of necrosis in solid tumors.

The increase in tumoral uptake of our necrosis probe after therapy is a parameter that will be explored in the future. For clinical translation, the timing after therapy is an important factor that should be evaluated as dying of cells after therapy is a dynamic process with ruminants of the dead cells getting cleared by the body. Moreover, parameters should be established for recognizing therapy efficacy by the hand of increased NACA uptake after therapy.

## 5. Conclusions

To conclude, we designed an indium-111 labeled NACA based on IRDye800CW containing an albumin-binding domain for detecting necrotic tumor tissue using SPECT/CT and FLI in tumor-bearing mice. The effect of doxorubicin therapy on MCF7-Luc2 tumors could be measured by an increased [^111^In]**3**-uptake in treated tumors over untreated tumors. The decreased [^18^F]FDG-uptake, a parameter for therapy efficacy assessment in the clinic, confirmed the therapy effect. Moreover, our data shows a potential of [^111^In]**3** detecting spontaneous necrosis in untreated tumors that can potentially indicate tumor aggressiveness and may provide valuable prognostic information. To our knowledge, this is the first time tumor necrosis is visualized by four in vivo imaging modalities in a single imaging session on the same animal (PET/SPECT/CT, FLI, and BLI). Future studies will be dedicated to elucidating the predictive power of [^111^In]**3**-uptake as a biomarker for tumor aggressiveness and as a tool to determine chemotherapy efficacy by measuring the net increase of [^111^In]**3**-uptake before and soon after the start of therapy. Detecting therapy-induced tumor necrosis may serve as an earlier marker for therapy efficacy than detecting morphological changes, currently used in the clinic.

## Data Availability

The data generated and/or analyzed during the current study are available from the corresponding author on reasonable request.

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
