# Peer review of "Improved Multimodal Tumor Necrosis Imaging with IRDye800CW-DOTA Conjugated to an Albumin-Binding Domain"

_cancers, 2022, doi:10.3390/cancers14040861_

Round 1

Reviewer 1 Report

The authors report on a novel multimodal probe (SPECT, NIR fluorescence) that selectively detects tumor necrosis, namely for early-stage determination of tumor aggressiveness and therapy efficacy. While the data is well presented, with appropriate controls in both in vitro and in vivo experiments, a few minor comments should be addressed or added before publication, which I believe could improve the manuscript :

  1. In the experimental part, on 2.13 (page 5), the authors mention that the data was corrected for the 18FDG decay, but no mention is done to 111In decay. Considering that the images and biodistribution experiments are done at different time points (6 to 72h) a correction for decay it becomes indispensable. As I assume that this has been done, I recommend adding this information, or removing the one mentioned exclusively to 18F. Providing that this was not the case, then all imaging and biodistribution need to be revised……
  2. In the protocol for the 111In labelling it is mentioned that DTPA was added, this is indeed a classical way how to complex any free metal in the solution, but no purification was performed further ? Are the HPLC chromatograms of pure fractions? If no purification was performed, for imaging at 24h p.i. I do not see how a presence of a 111In-DTPA would interfere (mainly due to its fast renal elimination), but for the 6h p.i. some of this later radiocomplex can still be detected… could the authors please comment on this?
  3. It seems that the elimination is rather renal, since no special uptake is detected in the liver, spleen or intestines that would justify a hepatobiliary route… maybe a comment in light of the lipophily of the probes/elimination route could be added ?
  4. Page 12, lines 376/377 : “received 3 lines of saline solution or dox treatment” should it be “times”? the sentence is strange otherwise
  5. Page 12, lines 383/384 : “uptake patterns within all tumors of both [111In]3 and [18F]FDG are heterogeneous and complementary” – is the heterogeneity mentioned within each cohort of animals? Or between the images obtained with each probe?

Author Response

The authors report on a novel multimodal probe (SPECT, NIR fluorescence) that selectively detects tumor necrosis, namely for early-stage determination of tumor aggressiveness and therapy efficacy. While the data is well presented, with appropriate controls in both in vitro and in vivo experiments, a few minor comments should be addressed or added before publication, which I believe could improve the manuscript:

  • Dear reviewer, many thanks for your time reading and commenting on our manuscript. We aimed to address your concerns as thoroughly as possible.

1. In the experimental part, on 2.13 (page 5), the authors mention that the data was corrected for the 18FDG decay, but no mention is done to 111In decay. Considering that the images and biodistribution experiments are done at different time points (6 to 72h) a correction for decay it becomes indispensable. As I assume that this has been done, I recommend adding this information, or removing the one mentioned exclusively to 18F. Providing that this was not the case, then all imaging and biodistribution need to be revised……

  • The decay correction mentioned in 2.13 refers to the decay correction applied in the reconstruction of the scan as the scan time is 30 minutes. These 30 minutes have a neglectable effect on the indium-111 (t1/2: 4032 min) but a considerable effect on the fluorine-18 (t1/2: 109.7 min). All mentioned organ/tumor uptake values are decay-corrected for the corresponding radionuclides and time after injection. We mention this now in line 220-223.

2. In the protocol for the 111In labelling it is mentioned that DTPA was added, this is indeed a classical way how to complex any free metal in the solution, but no purification was performed further? Are the HPLC chromatograms of pure fractions? If no purification was performed, for imaging at 24h p.i. I do not see how a presence of a 111In-DTPA would interfere (mainly due to its fast renal elimination), but for the 6h p.i. some of this later radiocomplex can still be detected… could the authors please comment on this?

  • The labeling reactions were performed with highly pure precursor material that allowed close to quantitative radiolabeling. As a result, no further purification was required after labeling. The presented HPLC-chromatograms in supplementary data 2 are the reaction mixtures as obtained without purification. The DTPA incorporates any unreacted indium-111. The incorporated indium is rapidly excreted by the body. In previous work we observed that radiometals incorporated with DTPA were only visible in the bladder at 4h p.i.. All experiments were performed with radiotracers that met the >95% radiochemical purity, ensuring minimal interference of any [111In]In-DTPA. If there would be any interference of the [111In]In-DTPA at 6 h p.i., this would at most be observed in the kidney uptake. We observed a kidney uptake pattern that peaked at around 24 h p.i.. This pattern is similar to other reported albumin binding complexes, see for example the work of Kuo and colleagues in reference 30. We included a reference in the manuscript to an older work from our group explaining the importance of adding DTPA at the end of the radiolabeling procedure, see reference 46.

3. It seems that the elimination is rather renal, since no special uptake is detected in the liver, spleen or intestines that would justify a hepatobiliary route… maybe a comment in light of the lipophily of the probes/elimination route could be added?

  • [111In]3 is still highly hydrophilic (LogD: -2.73), although less hydrophilic than [111In]1 (LogD: -3.58) and is therefore renally excreted. We now mention this in the discussion, line 456-458.

4. Page 12, lines 376/377 : “received 3 lines of saline solution or dox treatment” should it be “times”? the sentence is strange otherwise

  • Thanks to the reviewer for noticing the typing error. This sentence was indeed incorrectly formulated. We changed the sentence in line 379-381.

. Page 12, lines 383/384 : “uptake patterns within all tumors of both [111In]3 and [18F]FDG are heterogeneous and complementary” – is the heterogeneity mentioned within each cohort of animals? Or between the images obtained with each probe?

  • The mentioned heterogeneity refers to the heterogeneous uptake pattern within the tumor. The pattern is of the two emission tomography modalities are complementary to each other. We changed the formulation of the sentence in line 386-387.

Reviewer 2 Report

The paper reports on the development of radiolabeled cyanine dyes for imaging of tumor necrosis, which is of interest for a clinical application. These compounds hold the potential to serve as agents for in vivo imaging of tumor necrosis as a marker of tumor aggressiveness and for evaluating the efficacy of cancer therapies. Thus, the paper addresses a very timely topic in clinical cancer research.

Overall, the experiments have been thoroughly conducted and the drawn conclusions are comprehensible to the reader. Therefore, I recommend the paper to be published after minor revisions:

I would like to know whether the authors have any idea why compound 2 is in vivo so much less stable than compound 3. Is there a possible explanation with regard to the chemical structure? Maybe the authors could add one sentence in the paper with their theory on the different stabilities with regard to the chemical structure of 2 and 3.

In addition, there are some minor mistakes to be corrected:

- Write the relative configurations “L“ and “D“ of compounds using small caps. This is the case for D-luciferin e.g. in lines 187, 198, 364 and for L-Lys in lines 238, 240, 253, 260.

- In lines 450 and 468: change fluor-18 into fluorine-18

Author Response

The paper reports on the development of radiolabeled cyanine dyes for imaging of tumor necrosis, which is of interest for a clinical application. These compounds hold the potential to serve as agents for in vivo imaging of tumor necrosis as a marker of tumor aggressiveness and for evaluating the efficacy of cancer therapies. Thus, the paper addresses a very timely topic in clinical cancer research.

Overall, the experiments have been thoroughly conducted and the drawn conclusions are comprehensible to the reader. Therefore, I recommend the paper to be published after minor revisions:

  • Dear reviewer, many thanks for your time, reading and commenting on our manuscript. We aimed to address your concerns as thoroughly as possible.

I would like to know whether the authors have any idea why compound 2 is in vivo so much less stable than compound 3. Is there a possible explanation with regard to the chemical structure? Maybe the authors could add one sentence in the paper with their theory on the different stabilities with regard to the chemical structure of 2 and 3.

  • Regarding the stability of the tracers. A hypothetical explanation for the instability of compound 2 could be due to protease-mediated degradation of the amide bonds between the lysines. Since compound 3 was stable we could proceed our experiments and no further analysis was performed on the stability of compound 2.

In addition, there are some minor mistakes to be corrected:

- Write the relative configurations “L“ and “D“ of compounds using small caps. This is the case for D-luciferin e.g. in lines 187, 198, 364 and for L-Lys in lines 238, 240, 253, 260.

  • We decreased the font size of the annotations as suggested in lines 188, 199, 242, 244 257, 264, 368,

- In lines 450 and 468: change fluor-18 into fluorine-18

  • We addressed these mistakes as suggested in lines 462 and 480.